# *Gaidropsarus gallaeciae* (Gadiformes: Gaidropsaridae), a New Northeast Atlantic Rockling Fish, with Commentary on the Taxonomy of the Genus [note 1]

**DOI:** 10.3390/biology11060860

**Published:** 2022-06-03

**Authors:** Rafael Bañón, Francisco Baldó, Alberto Serrano, David Barros-García, Alejandro de Carlos

**Affiliations:** 1Servizo de Planificación, Consellería do Mar, Xunta de Galicia, Rúa dos Irmandiños s/n, 15701 Santiago de Compostela, Spain; 2Group de Estudo do Medio Mariño (GEMM), Edif. Club Naútico Bajop, 15960 Ribeira, Spain; 3Centro Oceanográfico de Cádiz (COCAD-IEO), CSIC, Puerto Pesquero, Muelle de Levante s/n, 11006 Cádiz, Spain; francisco.baldo@ieo.es; 4Centro Oceanográfico de Santander (COST-IEO), CSIC, Promontorio San Martín de Bajamar s/n, 39004 Santander, Spain; alberto.serrano@ieo.es; 5Centro Interdisciplinar de Investigação Marinha e Ambiental (CIIMAR/CIMAR), Terminal de Cruzeiros do Porto de Leixões, Avenida General Norton de Matos s/n, 4450-208 Matosinhos, Portugal; davbarros1985@gmail.com; 6Departamento de Bioquímica, Xenética e Inmunoloxía, Facultade de Bioloxía, Universidade de Vigo, Rúa Fonte das Abelleiras s/n, 36310 Vigo, Spain; adcarlos@uvigo.es

**Keywords:** Teleostei, taxonomy, rocklings, deep-sea

## Abstract

**Simple Summary:**

The genus *Gaidropsarus* is a poorly known group of marine fishes found from the intertidal zone to the deep sea in all three major oceans. The present taxonomic study describes a new deep-sea species of this genus originating from Galicia and Porcupine Banks, two seamount-like structures in the Northeast Atlantic. The results suggest that deep-water coral reefs could be an essential habitat for this species. The existence of this new species was previously flagged by the analysis of mitochondrial DNA sequences of the species of the genus described in the North Atlantic, and has been corroborated by morphological examination of the specimens.

**Abstract:**

A new species of rockling fish genus *Gaidropsarus* is described based on six specimens collected in Galicia and Porcupine Banks, in Atlantic European waters. An analysis of morphological characters has confirmed the specific status of specimens of a previously described clade by comparison of DNA sequences. *Gaidropsarus gallaeciae* sp. nov. it is distinguished from congeners by the following combination of characters: 43–44 vertebrae; 54–60 third dorsal fin rays; 44–52 anal fin rays; 21–23 pectoral fin rays; head length 21.1–25.2% of standard length (SL); length of the pelvic fin 16.2–19% SL; length of the first dorsal fin ray 15.8–27% of head length (%HL); eye diameter 15.8–20.5% HL; and interorbital space 21.7–28% HL. Using the nucleotide sequence of the 5’ end of the mitochondrial *COI* gene as a molecular marker, the genetic p-distance between the new species and its congeners far exceeds the usual 2%, granting the former the status of an independent taxon, which is in accordance with the morphological identification. A comparison with the other 12 valid species of the genus is presented. The study also highlights the morphological diversity resulting from the meristic and biometric variability of *Gaidropsarus* species and lays the groundwork for future taxonomic studies on this genus.

## 1. Introduction

The genus *Gaidropsarus* Rafinesque, 1810 shows a remarkable ecological diversity ranging from intertidal and near-shore to deeper areas up to 2000 m depth, and from arctic to temperate and subtropical waters. Species in this genus show an antitropical distribution, with most of them known only from the northern hemisphere, in the Atlantic Ocean, while species from the southern hemisphere produce a circumglobal ring of forms comprising the sub-Antarctic waters of the Atlantic, Indian and Pacific Oceans [1].

Following recent revisions [2,3] and the report of a new species [4], there are currently 12 valid species worldwide: *Gaidropsarus argentatus* (Reinhardt, 1837), *Gaidropsarus ensis* (Reinhardt, 1837), *Gaidropsarus granti* (Regan, 1903), *Gaidropsarus macrophthalmus* (Günther, 1867), *Gaidropsarus mediterraneus* (Linnaeus, 1758), *Gaidropsarus vulgaris* (Cloquet, 1824), *Gaidropsarus mauli* Biscoito & Saldanha 2018, *Gaidropsarus capensis* (Kaup, 1858) *Gaidropsarus insularum* Sivertsen, 1945, *Gaidropsarus novaezealandiae* (Hector, 1874) *Gaidropsarus pakhorukovi* Shcherbachev, 1995, and *Gaidropsarus parini* Svetovidov, 1986.

The classification of this genus is controversial, having been placed alternatively within the family Gaidropsaridae, Gadidae or Lotidae, although recent research that appears to be definitive includes it in the first group, forming its own family [5]. Fishes from this genus, commonly known as rocklings, are characterized by an elongated and relatively slender body, barbels present on the chin and at each anterior nostril on the snout, a first dorsal fin ray followed by a row of small fleshy filaments, a not indented anal fin and an uninterrupted lateral line is along its entire length [6].

The taxonomy of the genus *Gaidropsarus* is incomplete and inadequately known, due to the absence in the museums of any representative collections of the numerous species widely distributed in the World Ocean [1]. Further studies are required to evaluate the small differences between many of the described species, which at one time or another have been included in several nominal genera. Hence, a key and a complete list of species are not feasible at present [6]. Some recent publications based on molecular data of North Atlantic species highlighted several inconsistencies with existing morphology-based taxonomic concepts [2,7]. Moreover, the presence of probably undescribed species [2], new distribution records and extension ranges [8,9] and the finding of new species [4] confirm the limited knowledge of the genus.

Seamounts are hotspots of marine biodiversity with high species richness [10], but the composition and diversity of fish fauna on these habitats is not well documented. In addition to traditional fisheries sampling techniques, the use of non-invasive technologies such as remotely operated vehicles (ROVs), autonomous underwater vehicles (AUVs), and baited cameras (BCs) have improved knowledge of the seamounts fish fauna, including the discovery of potentially new species. However, specimen-based examination is needed to clarify their detailed taxonomy [11,12]. In fact, the report of new fish species found on seamounts is not uncommon [13,14].

The use of these “in vivo” sampling techniques has revealed the presence of unidentified rocklings at different deep-sea habitats of the NE Atlantic and the Mediterranean in association with the coral *Desmophyllum pertusum* (Linnaeus, 1758) or in cracks and crevices of chimneys [15,16,17].

The Galicia and the Porcupine Banks constitute two of the 557 seamount-like structures included in the limits of the OSPAR Convention (North-East Atlantic) [18]. The ichthyofauna of the Galicia Bank was studied from the 1990s onwards, resulting in a total of 139 catalogued species [8]. The Porcupine Bank has also been the subject of numerous ichthyological studies [19,20,21], but an updated species composition list has not yet been produced.

Molecular taxonomy is especially valuable for groups in which distinctive morphological features are difficult to observe or compare. During DNA analysis of a large sample of individuals of the genus *Gaidropsarus*, employing both mitochondrial (cytochrome oxidase subunit I, *COI*; cytochrome b, *CytB*; NADH dehydrogenase 2, *ND2*) and nuclear (Rhodopsine, *Rho*, Zic Family Member 1, *ZIC1*) markers, a set of sequences was revealed that differed significantly from those ascribed to recognized species [2,3]. This study provides a formal description of this set of individuals as a new species and explores the current state of knowledge of the genus.

## 2. Materials and Methods

Specimens come from two Atlantic seamount-like structures (Figure 1). The Galicia Bank is a non-volcanic seamount located off the northwest of the Iberian Peninsula, between 42°15′ N–43° N and 11°30′ W–12°15′ W, at water depths from 625 to 1800 m and at about 125 nautical miles from the coast, and is 50 km long in the E–W direction and 90 km on the N–S axis [8]. The presence of vulnerable species and habitats in this bank, such as *Lophelia* and *Madrepora* communities and black and bamboo coral aggregations were the basis for its inclusion in the Natura 2000 network as a Site of Community Importance [22]. The Porcupine Bank is located in the Northeast Atlantic, from 13° W to 15° W longitude and from 51° N to 54° N latitude, 200 km off the west coast of Ireland. It extends from a depth of 150 m to 4000 m of the abyssal plain, forming a seamount-like structure, which is connected by a narrow strip to the continental shelf. The closed water recirculation system in this area favours the retention of nutrients and plankton, creating an area of high productivity.

Sampling on the Galicia Bank was conducted aboard the R/V “Thalassa”, as part of the INDEMARES project (BanGal0810), an exploratory multi-gear survey, whereas sampling on the Porcupine Bank was conducted aboard the R/V “Vizconde de Eza” during the annual bottom-trawl surveys (Porcupine 2019), using a Baca-GAV 39/52 with a cod-end mesh size of 20 mm. Specimens were taken from the catch and frozen on board. In the laboratory, specimens were thawed, examined and photographed. With the exception of total length (TL) and standard length (SL), measurements are distances perpendicular to the length of the fish measured with a digital calliper to the nearest 0.1 mm on the type specimens and to nearest mm on the comparative material. Counts and measurements were recorded following Svetovidov [23,24]. All measurements are expressed as the percentage of standard length (%SL) or head length (%HL). Voucher specimens were deposited in the ichthyological collection of the Museo de Historia Natural, Universidade de Santiago de Compostela (MHNUSC).

To improve the knowledge of the natural variation of the species of the genus, a comprehensive review of the morphological characters, distribution and coloration of valid *Gaidropsarus* species reported in the ichthyological literature, mainly compiled in Svetovidov [23,24], Barros-García et al. [2] and Biscoito and Saldanha [4], was complemented with measurements and counts of own comparative material, when available.

A single *COI* sequence representative for each valid *Gaidropsarus* species available (*n* = 7) was retrieved from Genbank specimens belonging to the author’s project ‘Molecular identification of *Gaidropsarus* fishes’ (Code GSRUS) in BOLD systems (https://www.boldsystems.org/, accessed on 27 April 2022), including the sequence of the holotype of *Gaidropsarus gallaeciae* sp. nov. (KY250297). A sequence of *Notacanthus bonaparte* Risso, 1840 (KP845234) was used as outgroup.

These sequences were employed to construct a molecular cladogram using the Neighbour-Joining (NJ) method [25] in MEGA 11 [26] with confidence limits tested through a bootstrap procedure [27], after 2000 replicates.

## 3. Results

*Gaidropsarus gallaeciae* sp. nov.

http://zoobank.org/FA7C4849-4D40-45F1-9285-81B289ADE402 (accessed on 27 April 2022)

Figure 2, Figure 3 and Figure 4; Table 1

*Gaidropsarus* sp. 1: [2] (p. 26); *Gaidropsarus* sp.: [3].

### 3.1. Holotype

MHNUSC 10126-1 (Figure 2 and Figure 4), 111.4 mm TL, 98.5 mm SL, Galicia Bank, bottom trawl, 18 August 2010, 42.751 °N, −11.7713° W, 788.5 m; sample ID: ROL002; GenBank registration: KY250297.

### 3.2. Paratypes

MHNUSC 10126-2 (Figure 3), 88.2 mm TL, 76.4 mm SL, Galicia Bank, beam trawl; 19 August 2010, 42.4535° N, −11.4549° W, 788 m; sample ID: ROL001, GenBank registration: KY250298; MHNUSC-10126-3, 101.6 mm TL, 89.2 mm SL, Porcupine Bank, bottom trawl, 11 September 2019, 51.0316° N, −14.4061° W, 751 m; sample ID: ROL003, GenBank registration: MZ198255; MHNUSC-10126-4, 105.5 mm TL, 95.3 mm SL, Porcupine Bank, bottom trawl, 11 September 2019, 51.0316° N, −14.4061° W, 751 m; sample ID: ROL004, GenBank registration: MZ198256; MHNUSC-10126-5, 74.3 mm TL, 65.1 mm SL, Porcupine Bank, bottom trawl, 11 September 2019, 51.0316° N, −14.4061° W, 751 m; sample ID: ROL005, GenBank registration: MZ198257; MHNUSC-10126-6, 65.8 mm TL, 57.4 mm SL, Porcupine Bank, bottom trawl, 11 September 2019, 51.0316° N, −14.4061° W, 751 m; sample ID: ROL006, GenBank registration: MZ198258.

### 3.3. Comparative Material Examined 

Morphological data of the comparative material examined are shown in Appendix A.

*Gaidropsarus vulgaris* MHNUSC 25196-1, 320 mm TL, 8 February 2022, 42.3442, −8.9799, 62 m depth; MHNUSC 25196-2, 239 mm TL, 20 March 2014, 42.7937, −9.0212, 7 m depth; MHNUSC 25196-3, 295 mm TL, 16 January 2014, 42.3045, −8.8464, 7 m depth; MHNUSC 25196-4, 262 mm TL, 21 January 2014, 42.5003, −9.0499, 25 m depth; MHNUSC 25196-5, 258 mm TL 26 February 2014, 43.7598, −7.6387, 12 m depth.

*Gaidropsarus macrophthalmus* MHNUSC 25199-1, 201 mm TL, 10 September 2015, 52.1237, −13.8644, 405 m depth; MHNUSC 25199-2, 171 mm TL, 10 September 2015, 52.1237, −13.8644, 405 m depth; MHN USC 25199-3, 157 mm TL, 14 September 2017, 53.5014, −11.9862, 290 m depth; MHNUSC 25199-4, 185 mm TL, 24 September 2015, 52.0796, −13.0109, 729 m depth; MHNUSC 25199-5, 187 mm TL, 24 September 2015, 52.0796, −13.0109, 729 m depth.

*Gaidropsarus mediterraneus* MHNUSC 25195-1, 277 mm TL, 16 October 2015, 42.7346, −9.0906, 7 m depth; MHNUSC 25195-2, Madeira.

*Gaidropsarus ensis* MHNUSC 25197-1, 230 mm TL, MHNUSC 25197-2, 222 mm TL, 30 July 2015, 48.1953, −46.9417, 1004 m depth; MHNUSC 25197-3, 355 mm TL, 6 June 2015, 42.8497, −50.7967, 981 m depth; MHNUSC 25197-4, 245 mm TL, 30 July 2015, 48.1953, −46.9417, 1004 m depth.

*Gaidropsarus argentatus* MHNUSC-25198-1, 330 mm TL, 30 June 2015, 77.7206, −10.1256, 884 m depth; MHNUSC-25198-2, 282 mm TL, 30 June 2015, 77.1842, −11.4722, 657 m depth; MHNUSC-25198-3, 305 mm TL, 30 June 2015, 77.7206, −10.1256, 884 m depth.

### 3.4. Diagnosis

The new species belongs to the genus *Gaidropsarus* as defined by Iwamoto and Cohen [28] as having three dorsal fins barely separated from each other; the first with a single thickened unsegmented ray, the second with small, unsegmented rays in a fleshy ridge that rises within a groove and the third with segmented rays in an elongate fin, and five prominent individual barbels, four on the snout and one at the tip of the lower jaw. *Gaidropsarus gallaeciae* sp. nov. is morphologically distinct from all congeners by the following combination of characters: third dorsal-fin rays 54–60, anal-fin rays 44–52, pectoral fin rays 21–23, total vertebrae 43–44; anal fin base short, its length 39.6–48% SL; first dorsal fin ray moderately elongated, its length 15.8–27% HL and a wider interorbital space, 21.7–28% HL.

### 3.5. Differential Diagnosis

A detailed comparison between the *Gaidropsarus gallaeciae* sp. nov. and the other 12 valid congeners is provided in Table 2 and Table 3. According to our current knowledge of the genus, only three species, *G. macrophthalmus*, *G. capensis* and *G. granti*, share the low number of vertebrae found in the new species, while all other species have more than 45 vertebrae. It differs from *G. capensis* by having more third dorsal fin rays (54–60 vs. 43–52), anal fin rays (44–52 vs. 37–43) and pectoral fin rays (21–23 vs. 18–21), a wider interorbital space (21.7–28% HL vs. 13.5–19.5% HL), a different distribution area (NE Atlantic vs. SE Atlantic and SW Indian Oceans); from *G. macrophthalmus* by having more pectoral fin rays (21–23 vs. 17–22), a shorter anal fin base (39.6–48% SL vs. 48.5–50% SL), a longer pelvic fin (16.2–19% SL vs. 9.6–16.1% SL) and by the coloration pattern (uniform vs. mottled); and from *G. granti* by a wider interorbital space (21.7–28% HL vs. 10.5–17.6% HL), a longer first dorsal fin ray (15.8–27% HL vs. 12.7–14.9% HL) and by the coloration pattern (uniform vs. mottled).

*Gaidropsarus gallaeciae* sp. nov. differs from *G. novaezealandiae* by having fewer vertebrae (43–44 vs. 46–49), more pectoral fin rays (21–23 vs. 20–21), a wider Interorbital space (21.7–28% HL vs. 15.2–18.7% HL), a shorter anal fin base (39.6–48% SL vs. 48.2–51.5% SL), a deeper distribution range (751–788 m vs. 300–500 m) and a different geographical area (Northeast Atlantic vs. Southwest Pacific).

It differs from *G. insularum* by having fewer vertebrae (43–44 vs. 47–49), fewer third dorsal fin rays (54–60 vs. 66–70), fewer anal fin rays (44–52 vs. 50–57), a wider interorbital space (21.7–28% HL vs. 16.7–19.4% HL), a deeper distribution range (751–788 m vs. littoral) and a different geographical area (Northeast Atlantic vs. Southeast Atlantic).

It differs from *G. pakhorukovi* by having fewer vertebrae (43–44 vs. 46–47), fewer pectoral fin rays (21–23 vs. 22–26), a deeper distribution range (751–788 m vs. 670–1190 m) and a different geographical area (Northeast Atlantic vs. Southeast Atlantic).

It differs from *G. parini* by having fewer vertebrae (43–44 vs. 47–48), fewer third dorsal fin rays (54–60 vs. 60–64), fewer anal fin rays (44–52 vs. 52–53), fewer pectoral fin rays (21–23 vs. 23–25), a deeper distribution range (751–788 m vs. 310–680 m) and a different geographical area (Northeast Atlantic vs. Pacific Ocean: Nasca)

It differs from *G. mauli* by having fewer vertebrae (43–44 vs. 46–47), fewer pectoral fin rays (21–23 vs. 25–26), a shorter snout (20.9–25% HL vs. 25.4–26.7%HL), a shorter chin barbel (16.7–22.6% HL vs. 26.9% HL), a shorter pelvic fin (16.2–19% SL vs. 27.5–33.3% SL), a wider interorbital space (21.7–28% HL vs. 20.9–21.3% HL) and larger eyes (eye diameter 15.8–20.5% HL vs. 10.4–12% HL).

It differs from *G. vulgaris* by having fewer vertebrae (43–44 vs. 46–49), a wider interorbital space (21.7–28% HL vs. 14.4–19.5% HL), a deeper distribution range (751–788 vs. 10–120) and by the coloration pattern (uniform vs. spotted).

It differs from *G. mediterraneus* by having fewer vertebrae (43–44 vs. 46–50), more pectoral fin rays (21–23 vs. 15–19), a longer pelvic fin (16.2–19%SL vs. 13–15.5%SL), a deeper distribution range (751–788 vs. 0–40) and different coloration (pinkish vs. brown).

It differs from *G. argentatus* by having fewer vertebrae (43–44 vs. 49–53), a larger anal fin base (39.6–48 vs. 38.7–39.8), a shorter snout (20.9–25% HL vs. 25.2–27%HL) and a shorter first dorsal fin ray (15.8–27 % HL vs. 24.1–43% HL).

It differs from *G. ensis* by having fewer vertebrae (43–44 vs. 50–54), fewer gill rakers (6–9 vs. 12–14 in the inner row) and a shorter first dorsal fin ray (15.8–27% HL vs. 82.1–145.5% HL).

### 3.6. Etymology

The name gallaeciae derives from the latin Gallaecia, an ancient Roman Iberian province, now called Galicia, the westernmost region of Spain, in reference to the name of the Galicia Bank where the holotype was collected.

### 3.7. Description

Counts and measurements of type specimens are shown in Table 1. Body elongate and relatively slender, maximum body depth is contained from 5 to 6.4 times in SL; moderate and round eyes, horizontal eye diameter 1 to 1.5 times in snout length; snout short and rounded, its length 4.1 to 4.8 times in head length; mouth large (Figure 4A), slightly oblique, reaching a vertical through the posterior margin of orbit; upper jaw slightly protruding beyond lower jaw; first dorsal fin short, contained 3.7 to 6.3 times in HL; a small anterior nostril placed near de base of the barbel; posterior nostril oval, large, close to orbit; barbel present on chin, its length approximately equal to eye diameter and 1 to 1.4 times in snout lenght, and one barbel at each anterior nostril on the snout; first dorsal fin ray elongated, followed by a second dorsal fin of short fleshy filaments. The dentition consists of densely packed bands of small conical elements in both jaws (Figure 4B,D); the outermost row in the upper jaw and the innermost row in the lower jaw are larger; fang-like teeth absent in both jaws; conical teeth on the vomer boomerang-formed (Figure 4C); palatine teeth absent; gillrakers in the form of dentated tubercles (Figure 4F), 1 + 6 – 8 on the outer side of the first arch and 1+5–8 on the inner side. The colouration of fresh specimens is pinkish-reddish on the head, body and fins, and greyish on the ventral visceral part (Figure 2 and Figure 3).

### 3.8. Molecular Taxonomic Remarks

Figure 5 shows a molecular cladogram of valid *COI Gaidropsarus* species sequences publicly available. In this figure, *Gaidropsarus gallaeciae* sp. nov. is located at an independent and well differentiated branch. The genetic distance of the sequence of the new species from those of its congeners far exceeds 2%, which is the cut-off value for species delimitation in teleost marine fishes [29]. This figure partially illustrates previously obtained results, in which this species is reported as *Gaidropsarus* sp. 1 [2] or as *Gaidropsarus* sp. [3].

### 3.9. Habitat and Distribution

Known specimens were collected from two seamount-like structures in the Northeast Atlantic, the Galicia and Porcupine Banks, at 788 and 751 m depth, respectively (Figure 1). All specimens were caught together with a large amount of live and dead cold-water coral of *D. pertusum*, *Desmophyllum dianthus* (Esper, 1794) and *Madeprora oculata* Linnaeus, 1758, a fact that support this being the preferred habitat of the species. In the Galicia Bank, this habitat was named as “Summit Sands with CW coral reef patches“, corresponding with A6.611 Deep-sea *D. pertusum* reefs in the EUNIS classification [22]. Considering the presence of cold-water coral communities around the world, the new species it is likely to be widely distributed, but most probably throughout the eastern Atlantic and Mediterranean areas.

### 3.10. Accompanying Fauna

The two specimens of the Galicia Bank were collected along with 21 other fish species, including several gadiform such as *Guttigadus latifrons* (Holt & Byrne, 1908), *Halargyreus johnsonii* Günther, 1862, *Lepidion lepidion* (Risso, 1810), *Mora moro* (Risso, 1810) and *Nezumia aequalis* (Günther, 1878). The list of invertebrates caught included around 70 different species of crustaceans, molluscs, echinoderms and cnidarians (dead coral, *D*. *pertusum*, *M. oculata*).

The four specimens of the Porcupine Bank were collected along with one another rockling species, *G. granti* [9], and 33 other fish species, including also several gadiform fishes such as *Trachyrincus scabrus* (Rafinesque, 1810), *L. lepidion*, *Phycis blennoides* (Brünnich, 1768) or *H. johnsonii* among others. The list of bottom living invertebrates collected from the same site included 38 species of crustaceans, molluscs, echinoderms and cnidarians (dead coral, *D. pertusum*, *D*. *dianthus*).

## 4. Discussion

The morphology of the genus *Gaidropsarus* is conservative, making it difficult to find diagnostic characters and to establish an identification key for all known species. *Gaidropsarus gallaeciae* sp. nov. shares many morphological characters with the other congeneric species. A combination of meristic, biometric, colouration, geographical distribution and depth characters is therefore needed to differentiate the new species from all congeneric species.

The number of vertebrae is an important diagnostic character in distinguishing species of *Gaidropsarus*. On this basis, the species of this genus can be divided into two groups, either those that may have 45 vertebrae or less or those with more than 45 vertebrae. Our newly described species, *Gaidropsarus gallaeciae* sp. nov. is included in the first group together with *G. macrophthalmus*, *G. capensis* and *G. granti*.

Traditional taxonomy is descriptive, but the diagnostic characters of many hitherto unrevised fishes come from early manuscripts, which often refer to the examination of only a few specimens, and these results have come down to the present day with minimal changes. However, the magnitude of the variation of morphological characters in fishes, mainly biometrics and meristics, is not properly known and they are often underestimated, being the cause of erroneous denominations and the emergence of synonymies [30]. This seems to be the case for *Gaidropsarus* species. For example, *G. gutattus*, now considered a synonym of *G. mediterraneus* [2,3], was originally described by Collett [31] based on two specimens and only three more were subsequently analysed [23]. Therefore, the morphology of this species has only been based on the examination of five specimens, which is clearly insufficient to know the natural morphological variation of a species. The number of specimens examined was also low for the rest of the species. The specific distinction of *G. insularum*, *G. novaezealandiae* and *G. parini* carried out by Svetovidov [23] is based on only three, ten and two specimens, respectively, so it is not surprising that the valid status of these species has been questioned by Andrew et al. [32].

The main distinctive characters found in *Gaidropsarus gallaeciae* sp. nov. were the aforementioned number of vertebrae, the interorbital space, the length of the anal fin base, the length of the first dorsal fin ray and the length of the chin barbel. However, given the small number of specimens of *Gaidropsaurus* species examined, the number of distinctive characters may be reduced in the future.

Among the descriptive characters, the meristic ones have traditionally been used in the identification keys of *Gaidropsarus*. In fact, the number of fin rays is an important feature for taxonomic discrimination between species of this genus [23,24]. Short, non-overlapping ranges of morphological characters will favour the detection of distinctive characters, while wide, overlapping ranges make it difficult. The main diagnostic characters of the genus were compiled by Svetovidov [23,24] and successive updates [2,4]. In their revision, Barros-García et al. [2] show a large interspecific overlap of the meristic features, resulting in a set of conservative morphological traits. On the other hand, only a few other characters were used as diagnostics. The length of the first dorsal fin ray is longer in the boreal *G. ensis* and *G. argentatus* with respect to other species, whereas *G. macrophthalmus* is distinguished by the presence of enlarged canine teeth on the upper jaw [23,24,28]. Small eyes, contained five or more times in the head length separate *G. vulgaris*, *G. granti* and *G. mediterraneus* from *G. macrophthalmus*, with large eyes, contained less than five times in the head length [28]. However, measurements of comparative material of *G. macrophthalmus* show the eye diameter is contained 4.9 to 6.3 times in the head length, which refutes this character as diagnostic. Further sampling effort would be needed to gain knowledge of the true morphological variation of *Gaidropsarus* species in order to create more reliable identification keys.

The coloration pattern is also a diagnostic character in the genus *Gaidropsarus* [4,23,24,28]. The coloration of *G. granti*, with a whitish sinuous longitudinal band is a quick and useful diagnostic character for the species [9], and the presence of dark spots on the dorsal parts of head and body, and on the second dorsal and caudal fins is also a diagnostic character for *G. vulgaris* [28]. The colour pattern of this genus varies with depth, from a polymorphic colour in shallower species to a uniform coloration in the deeper ones, most likely tied to how light penetrate the ocean water and camouflaging is needed. Thus, *G. mediterraneus*, the shallower species of the genus, shows a cryptic and variable coloration, which has probably led to the consideration of two different species, *G. mediterraneus* from the continental area and *G. gutattus* from the insular one, when, in fact, they are one and the same [2,3]. Whereas *G. mediterraneus* has a more or less uniform brown colouration, *G. gutattus* exhibits a whitish mottled one. This synonymy was already pointed out by Orsi Relini & Relini [33], who reported that *G. mediterraneus* sometimes showed the typical coloration of *G. gutattus*, with irregular light spots on its dorsal and lateral dark brown surfaces. This is also the most probable cause of the erroneous record of *G. guttatus* in continental area [34].

In contrast, the North Atlantic deeper species such as *G. granti*, *G. argentatus*, *G. mauli* and *Gairopsarus gallaeciae* sp. nov. show a similar homogeneous pink-reddish coloration which could difficult their correct identification, particularly in juvenile stages.

The analyses of Barros-García et al. [2,3] seems to point to the real composition of species of this genus, with a reduction of valid shallower species, due to synonymy recognition, suggesting, furthermore, the existence of a greater diversity hidden in the deep. The recent record of *G. mauli* [4] and *Gaidropsarus gallaeciae* sp. nov. themselves would confirm this hypothesis. Moreover, according to the molecular results, other deep-sea species remain yet undescribed [2]. Considering that most of the deep-sea areas are unexplored and that the occurrence of *Gaidropsarus* species reported by in vivo sampling techniques (ROVs, AUVs, BCs) is not unusual, an increased number of undiscovered deep-sea species of this genus is predictable.

Eggs, larvae and juveniles of *Gaidropsarus* species are pelagic [6]. Pelagic early stages and the absence of physical barriers in the ocean should prevent rapid speciation events, but this statement contrasts with the fact of finding greater diversity at depth. However, depth-related ecological niche axis along which divergence occurs is due to local adaptation to diverse feeding habitats, light conditions, spawning sites, or other ecological factors [35]. For instance, speciation in the Pacific rockfish genus *Sebastes* is associated with divergence in habitat depth and a depth-associated morphology, in the absence of geographic barriers [36].

Therefore, diversification processes according to deep-sea environments is proposed for the genus *Gaidropsarus*. Ecological speciation occurs when adaptation to different environments or resources causes reproductive isolation [37]. Changing habitats with depth can create ecological opportunities for colonisation processes, promoting species diversification. The recently discovered species *G. mauli* was first found in a hydrothermal vent site, in crevices along rocky walls in the vicinity of the venting fluids [4]. Given the abundant presence of live and dead coral in the catches of all of the specimens, the occurrence of *Gaidropsarus gallaeciae* sp. nov. could be associated with the presence of cold-water coral reefs, as also occurs with *G. granti* [9]. In fact, both species were caught in the same haul in the Porcupine Bank, and both were also found in samples with coral in the Galicia Bank [38] which reinforces this likely coexistence and niche overlap. Several other fish species caught alongside the new species such as *M. moro*, *L. lepidion* and *G. latifrons* are also associated with the occurrence of cold-water corals [39], reinforcing the above. Association between *Gaidropsarus* species and cold-water corals has been often observed in live specimens on and between live and dead coral thickets [12,17,39,40]. Underwater observations have also shown much imagery evidence of a strict territorial behaviour of *Gaidropsarus* sp. in *D. pertusum* or *M. oculata* colonies [41]. The size of *Gaidropsarus gallaeciae* sp. nov. appears to be small, up to 11 cm TL, which could also be a morphological adaptation to shelter among the branches of corals as protection against predators. Hydrothermal vent fields and cold-water corals are two of the varied habitats responsible for the high biodiversity found in the deep ocean [42].

DNA barcoding greatly facilitates the grouping of individuals into putative species, which must then be validated through morphological scrutiny by taxonomic experts. Though morphology is the traditional technique used in alpha taxonomy, genetic tools are becoming increasingly common in studies describing new species, especially when morphological data are ambiguous [43]. In this aspect, Renner et al. [44] recommended DNA-based diagnoses of new species in all taxonomic groups, not just bacterial. A DNA barcoding analysis including the calculation of *COI* genetic distances [2] clearly differentiated *Gaidropsarus gallaeciae* sp. nov. (then reported as *Gaidropsarus* sp. 1) from six valid and two unidentified species from the North Atlantic. Considering that this previous study was based only on mitochondrial DNA sequences, a multilocus species delimitation analysis was carried out, including both mitochondrial (*COI*, *CytB*, *ND2*) and nuclear (*Rho*, *ZIC1*) genetic markers, that finally confirmed this previous finding [3].

Although no southern hemisphere species sequences are available, a BOLD search of *Gaidropsarus gallaeciae* sp. nov. returns a difference of 4.11%, a typical species differentiation distance, with a private *COI* sequence assigned to southern species *G. novaezealandiae*. Interestingly, this would be the smallest genetic distance found between the new species and all other congeneric species, as this species is more distantly related to all of the North Atlantic species examined, ranging from 13.21 to 17.36% [2].

A revision of this genus based on extensive collections of specimens and DNA sequencing is needed [4]. Therefore, it is necessary to complete the sequence database with specimens from southern hemisphere to better understand the interspecific relationships of this genus. Without the slightest doubt, the integrative study of traditional and molecular taxonomy can highlight identification mistakes and incongruities between the two disciplines, helping to reveal cryptic species, to identify immature specimens, and to clarify synonymies [45].

## 5. Conclusions

The occurrence of a new fish species *Gaidropsarus gallaeciae* sp. nov. is well supported by morphological and molecular analyses that clearly differentiate it from other known species. The taxonomy of the genus *Gaidropsarus* remains poorly understood. The relatively large number of species in this genus, the scattered distribution of many of them, and the small number of specimens examined and/or found are probably the main reasons for this insufficient knowledge. Although morphological characters are conservative, overlapping to a large extent between species, some diagnostic characters can be identified. These, together with colouration, geographical distribution and depth range can currently be applied for species identification. Molecular taxonomy of North Atlantic species has helped to resolve some taxonomic inaccuracies, but also flags the presence of undescribed species. Examination of more specimens and obtaining DNA sequences of southern hemisphere species should be the next objective to clarify the taxonomy of this difficult group.

## Figures and Tables

**Figure 1 biology-11-00860-f001:**
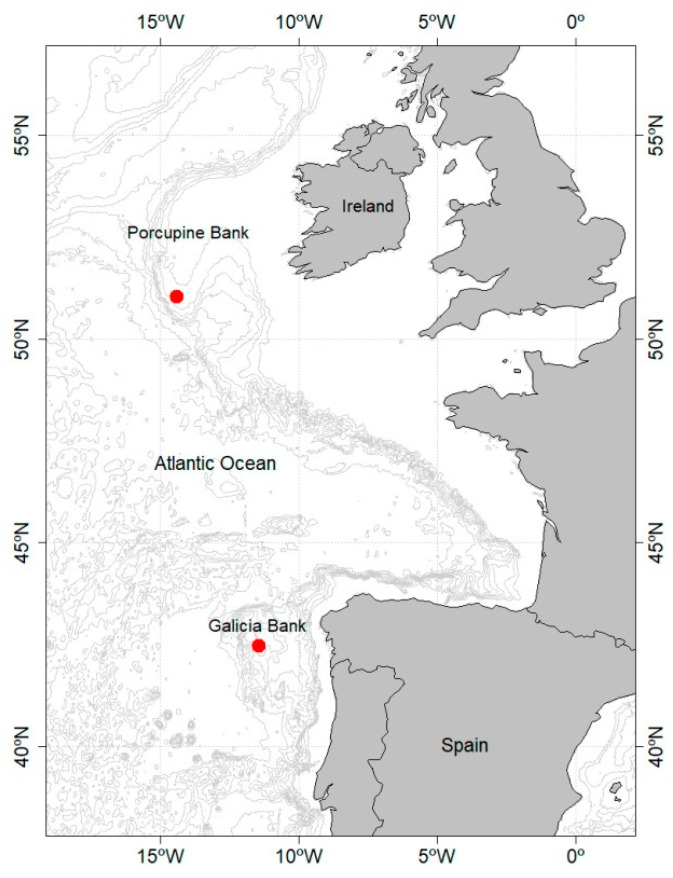
Map showing the location of catches of *Gaidropsarus gallaeciae* sp. nov. specimens in the Galicia and Porcupine Banks, in the northeast Atlantic.

**Figure 2 biology-11-00860-f002:**
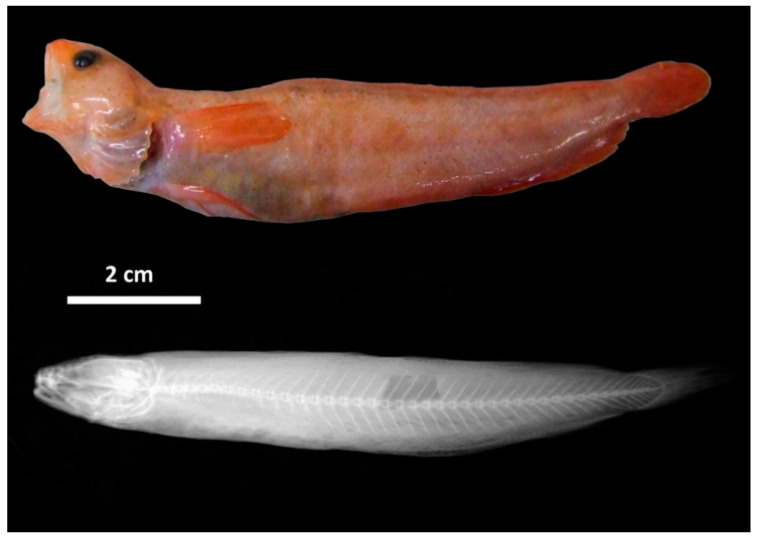
*Gaidropsarus gallaeciae* sp. nov., holotype (MHNUSC 10126-1) 98.5 mm SL (**top**), X-ray of entire body (**down**).

**Figure 3 biology-11-00860-f003:**
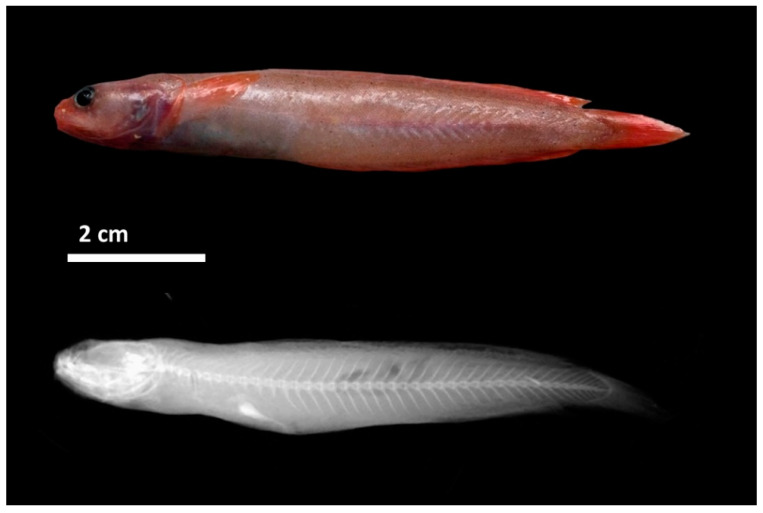
*Gaidropsarus gallaeciae* sp. nov., paratype (MHNUSC 10126-2) 76.4 mm SL (**top**), X-ray of entire body (**down**).

**Figure 4 biology-11-00860-f004:**
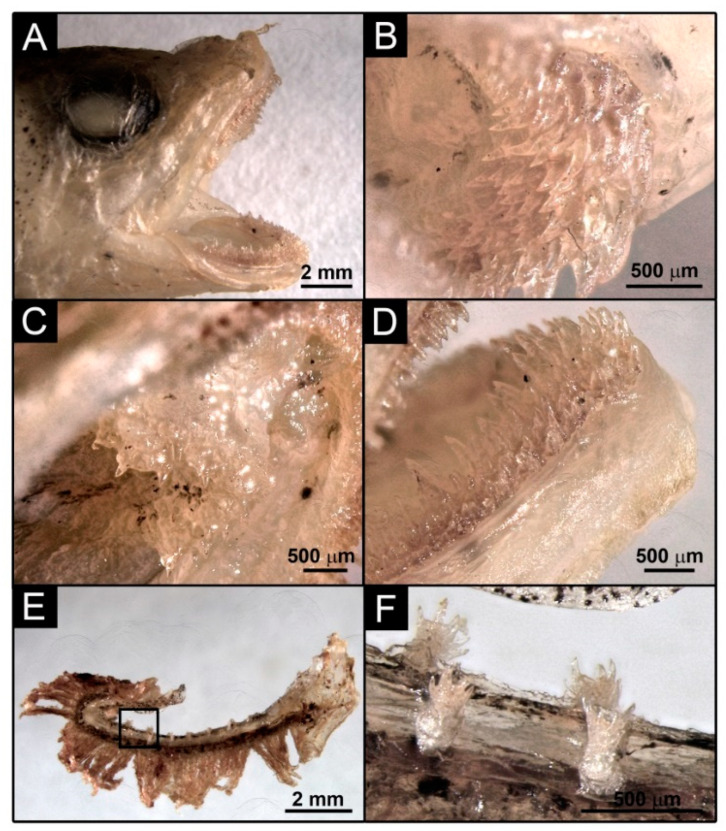
Details of the head structures of the holotype of *Gaidropsarus gallaeciae* sp. nov.: (**A**) mouth; (**B**) upper jaw teeth; (**C**) vomer teeth; (**D**) lower jaw teeth; (**E**) first gill arch; (**F**) gill rakers.

**Figure 5 biology-11-00860-f005:**
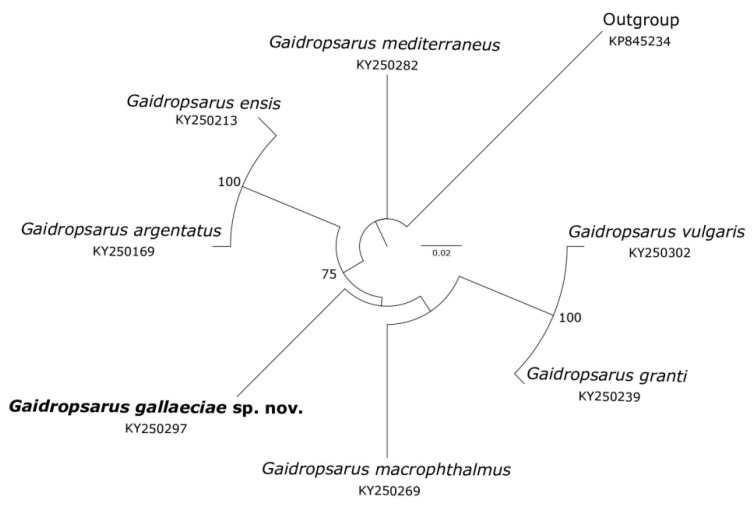
Neighbour-Joining tree, based on p-distances, for COI sequences of *Gaidropsarus* analyzed in this study. Numbers at the main nodes are bootstrap percentages after 2000 replicates. Only values higher than 70% are shown. The scale refers to the number of substitutions per nucleotide.

**Table 1 biology-11-00860-t001:** Morphological data of the type material of *Gaidropsarus gallaeciae* sp. nov.

	Holotype			Paratypes			Mean ± SE
TL	111.4	88.2	101.6	105.5	74.3	65.8	91.1 ± 7.4
SL	98.5	76.4	89.2	95.3	65.1	57.4	80.3 ± 6.8
As % SL							
Head length	22.4	21.1	25.2	22.9	24	23.9	23.3 ± 0.6
1st Predorsal length	21.8	27.7	22.6	22.4	23	24.7	23.7 ± 0.9
3rd Predorsal length	37.9	34.4	36.4	33.7	40.4	38	36.8 ± 1
2nd dorsal base fin length	11.7	9.8	11.3	10.3	11.4	11.7	11 ± 0.3
3rd dorsal base fin length	56.4	59.7	64.4	59	60.1	55.9	59.3 ± 1.2
Anal base fin length	46.4	48	47.2	46.6	39.6	45.6	45.6 ± 1.2
Pectoral fin length	15.6	16.1	17.5	16.1	16.9	15.3	16.3 ± 0.3
Pelvic fin length	16.2	18.8	17.8	16.4	18	19	17.7 ± 0.5
Preanal length	48.9	45.9	48.4	43.4	47.2	49.1	47.2 ± 0.9
Body depth	19.9	15.7	19.4	21.6	19.7	19.5	19.3 ± 0.8
Prepectoral length	22.2	22	22.8	22	24.1	27.5	23.4 ± 0.9
Prepelvic length	17.9	18.6	19.3	16.8	20.4	21.3	19.1 ± 0.7
Caudal peduncle height	6.3	6.2	7.4	6.9	7.2	7.7	7 ± 0.2
As % HL							
Snout length	24	21.1	20.9	24.3	25	22.6	23 ± 0.7
Eye diameter	15.8	20.5	19.6	19.7	18.6	17.5	18.6 ± 0.7
Postorbital length	64.3	58.4	54.7	56	56.4	60.6	58.4 ± 1.5
Interorbital space	24.4	21.7	23.1	28	25	26.3	24.8 ± 0.9
Upper jaw length	37.9	44.7	39.6	47.2	41	43.1	42.3 ± 1.4
Lower jaw length	34.4	37.3	32	36.7	36.5	40.1	36.2 ± 1.1
Chin barbel length	16.7	16.8	18.6	21.1	21.8	22.6	19.6 ± 1.1
1st dorsal fin ray length	15.8	19.3	24	22	26.3	27	22.4 ± 1.8
Meristic							
3rd dorsal fin rays	60	57	57	58	54	58	57.3 ± 0.8
Anal fin rays	44	52	46	49	50	50	48.5 ± 1.2
Pectoral fin rays	22	23	21	22	23	22	22.2 ± 0.3
Ventral fin rays	7	7	7	7	7	7	7
Gill rakers (inner)	1 + 8	1 + 5	1 + 6	1 + 6	1 + 8	1 + 6	7.5 ± 0.5
Gill rakers (outer)	1 + 8	1 + 7	1 + 7	1 + 8	1 + 6	1 + 7	8.2 ± 0.3
Vertebrae	44	43	–	–	–	–	43.5 ± 0.3

**Table 2 biology-11-00860-t002:** Comparison of morphological data between *Gaidropsarus gallaeciae* sp. nov and 12 valid species of the genus. Abbreviations are as follow: *Gaidropsarus gallaecia* sp. nov. (GGAL), *G. mauli* (GMAU), *G. argentatus* (GARG), *G. ensis* (GENS), *G. mediterraneus* (GMED), *G. vulgaris* (GVUL), *G. granti* (GGRA), *G. macrophthalmus* (GMAC), *G. insularum* (GINS), *G. novaezealandiae* (GNOV), *G. capensis* (GCAP), *G. pakhorukovi* (GPAK), and *G. parini* (GPAR).

Species	GGAL	GMAU	GARG	GENS	GMED
As % SL					
Head length	21.1–25.2	23.4–25.4	19.7–25.1	19–22	18.8–24
1st Predorsal length	21.8–27.7	22.8–23.9	20.7–22.6	18.7–20.2	18.6–18.9
3rd Predorsal length	33.7–40.4	36–36.3	31.7–36.8	29.1–32.3	20.1–38.5
2nd dorsal base fin length	9.8–11.7	—	8.6–11.4	8–11.3	13.2–18.4
3rd dorsal base fin length	55.9–64.4	—	57.1–62.4	59.3–64.4	54.1–60.9
Anal base fin length	39.6–48	—	38.7–39.8	39.9–46.3	45–52.2
Pectoral fin length	15.3–17.5	17.8–19.4	16.1–18.9	17–20	12.3–14.6
Pelvic fin length	16.2–19	27.5–33.3	18.1–21.5	17–26.3	13–15.5
Preanal length	43.4–49.1	50.8–53.1	51.4–53.4	48–50	44.1–51.1
Body depth	15.7–21.6	15.2–21.9	15.6–23.5	16.7–25.2	14–19.3
Prepectoral length	22–27.5	—	20.7–28	17.8–21.3	20.5–22.7
Prepelvic length	16.8–21.3	—	16.9–19.9	12.7–16	15–17.2
Caudal peduncle height	6.2–7.7	5.6–6.8	5.5–7.4	5.1–7.2	4.5–6.3
As% HL					
Snout length	20.9–25	25.4–26.7	25.2–27	23.6–27.9	18.8–30.4
Eye diameter	15.8–20.5	10.4–12	14.8–21.8	17.3–24.5	13.3–22.5
Postorbital length	54.7–64.3	—	57.3–58.4	54.1–59.1	60.63.5
Interorbital space	21.7–28	20.9–21.3	13.1–23.1	14.4–25.1	9.1–25.7
Upper jaw length	37.9–47.2	—	44.7–47.7	45.3–64.8	42.9–45.6
Lower jaw length	32–40.1	—	36.6–41.1	36.1–60.3	38.9–40.2
Chin barbel length	16.7–22.6	26.9	19.8–23.8	15.1–20.8	15.3–18.5
1st dorsal fin ray length	15.8–27	21.3–25.4	24.1–43	82.1–145.5	14.9–42
Meristic					
3rd dorsal fin rays	54–60	57–58	52–65	52–64	48–63
Anal fin rays	44–52	46–47	43–51	40–48	41–53
Pectoral fin rays	21–23	25–26	22–25	20–27	15–19
Ventral fin rays	7	9	7–8	6–7	5–8
Gill rakers (inner)	6–9	9	10–11	12–14	9–11
Gill rakers (outer)	7–9	7–8	8–11	11–13	7–10
Vertebrae	43–44	47–48	49–53	50–54	46–50
**Species**	**GVUL**	**GGRA**	**GMAC**	**GINS**	**GNOV**
As % SL					
Head length	23.6–25.9	20.9–25.5	19.3–23.2	18.7–21.5	17.9–20.7
1st Predorsal length	22.1–24	—	20.5–22.9	—	—
3rd Predorsal length	36.4–38.1	21.1–37.9	33.3–38.3	—	—
2nd dorsal base fin length	11.3–13.9	10.7	8.6–11.7	8.5–9.8	—
3rd dorsal base fin length	54.9–61.1	54.4–59.7	55.6–66.3	65.1–67.5	58.5–65.3
Anal base fin length	40.5–45.3	43.6–45.6	48.5–50	46.4–49.6	48.2–51.5
Pectoral fin length	14.1–15.4	13.8–15.4	14.7–15.5	—	—
Pelvic fin length	17.4–20.3	15.5–23.1	9.6–16.1	—	—
Preanal length	48.9–54.8	48.7–54.8	44–47.6	—	—
Body depth	14.8–20.4	13.1–14	14.2–19.5	—	—
Prepectoral length	23.5–25.4	—	19.3–25	—	—
Prepelvic length	18.6–20.7	—	16.3–20.2	—	—
Caudal peduncle height	7–8.5	5.6–6.9	4.8–7.1	6.8–8.5	6.3–8.1
As % HL					
Snout length	21.2–26.6	19.3–29.2	21–26	27.6	—
Eye diameter	10.5–16.7	13.7–18.8	16–23.7	—	15.2–19
postorbital length	55.3–65.2	59.6	54.3–59.1	—	
Interorbital space	14.4–19.5	10.5–17.6	12.5–26.5	16.7–19.4	15.2–18.7
Upper jaw length	42.3–49.3	42.9	46.2–52.9	59.2–61.3	—
Lower jaw length	36.8–40	41.7	36.6–44	48.4–55.1	—
Barbel length	19.6–24.2	—	14–22.2	—	—
1st dorsal ray length	9.5–16.9	12.7–14.9	10.1–25.1	11.2–25	20–27.9
Meristic					
3rd dorsal fin rays	56–64	55–60	48–59	66–70	56–69
Anal fin rays	46–54	45–52	40–50	50–57	50–59
Pectoral fin rays	20–24	20–22	17–22	19–22	20–21
Ventral fin rays	6–7	7–8	6–7	—	7–8 (5)
Gillraker (inner)	10–11	—	8–11	9	9–10
Gillraker (outer)	7–9	10	6–9	7	6–8
Vertebrae	46–49	44–47	43–47	47–49	46–49
**Species**	**GCAP**		**GPAK**	**GPAR**	
As % SL					
Head length	19.4–22.5		23.7–24.7	22.1–22.8	
1st Predorsal length	—		24.4–25.3	17.8–18.5	
3rd Predorsal length	—		—	—	
2nd dorsal base fin length	12.2–13.2		12.8–15.5	10.4–11.6	
3rd dorsal base fin length	—		55.3	56–58.1	
Anal base fin length	48.4–49		41.7	43.8–48.6	
Pectoral fin length	—		17.7–19.2	17.3–17.8	
Pelvic fin length	—		—	19.9–20.7	
Preanal length	—		—	45.2–48.5	
Body depth	16.5–17.3		—	—	
Prepectoral length	—		—	—	
Prepelvic length	—		22.4	—	
Caudal peduncle height	7–8.1		6.5	6.7–7.1	
As % HL					
Snout length	28.2–33.2		—	—	
Eye diameter	16.1–20.9		17.2–19.8	13.9–16.4	
postorbital length			—	—	
Interorbital space	13.5–19.5		16	—	
Upper jaw length	48.8–52.1		—	—	
Lower jaw length	—		—	—	
Barbel length	—		—	—	
1st dorsal ray length	19.5–32.5		12–15.1	26.7–28	
Meristic					
3rd dorsal fin rays	43–52		60–62	60–64	
Anal fin rays	37–43		50–51	52–53	
Pectoral fin rays	18–21		22–26	23–25	
Ventral fin rays	6–7		7–8	7–8	
Gillraker (inner)	8–9		9	10	
Gillraker (outer)	4–9		9	7	
Vertebrae	41–43		46–47	47–48	

**Table 3 biology-11-00860-t003:** Comparison of coloration, geographical and depth distributions between *Gaidropsarus gallaeciae* sp. nov and 12 valid species of the genus.

Species	Coloration	Distribution
*Gaidropsarus gallaeciae* sp. nov.	Pinkish overall, greyish on the abdominal region	NE Atlantic: Galicia Bank and Porcupine Bank, 751–788 m depth
*Gaidropsarus* *mauli*	Pinkish overall, less intense on the abdominal region, varying from more or less uniform to a mottled pattern	Atlantic Ocean: Azores and Bay of Biscay; 850–1700 m depth
*Gaidropsarus argentatus*	Uniform reddish-brown to light brick red; pink ventrally	Arctic and Atlantic Oceans, from west Spitzbergen, Norwegian Sea, Iceland, south Greenland and off Labrador; 150–2260 m depth
*Gaidropsarus ensis*	Light brick red, belly tinted red with blue-grey tinge	N Atlantic: Off Newfoundland and Labrador and west of British Isles; 600–1500 m depth
*Gaidropsarus mediterraneus*	Varied, back and upper flank brown, sometimes reddish brown, grading to pale brown-white on the ventral, with pale spots along the sides; blackish with white blotches mainly in the Macaronesian specimens	NE Atlantic, from Norway and British Isles south to Morocco, including Canaries, Azores and Madeira, and Mediterranean; 0–40 m depth
*Gaidropsarus* *vulgaris*	Pale cream to pink or reddish with chocolate brown spots on head and body	NE Atlantic, from Norway and Iceland south to Gibraltar, including Madeira and Mediterranean; 10–120 m depth
*Gaidropsarus* *granti*	Back brown, with irregular brown creamy blotches and spots and a whitish longitudinal sinuous band along upper flank	NE Atlantic, in Porcupine Bank (southwest of Ireland); Galicia Bank off Spain; Azores, Madeira and Canary Islands and Mediterranean; 20–823 m depth
*Gaidropsarus macrophthalmus*	Back mottled deep brown, flanks reddish, belly pink	NE Atlantic, from Faroe Islands and British Isles to south of the Azores and Mediterranean; 150–600 m depth
*Gaidropsarus insularum*	Chocolate-brown	SE Atlantic: southern tip of Africa, Tristan da Cunha and Gough islands; SW Indic: St. Paul and Amsterdam Islands; littoral
*Gaidropsarus novaezealandiae*	Head, body and fins dark reddish brown, purplish grey ventrally	SW Pacific: New Zealand and south of Tasmania; 0–50 m, but two specimens collected at 300–500 m
*Gaidropsarus* *capensis*	Unknown in live fish	SE Atlantic and SW Indian Oceans: southern Africa; to 45 m depth
*Gaidropsarus pakhorukovi*	Brownish-grey, darker dorsally	SW Atlantic: Rio Grande Seamount; 670–1190 m depth
*Gaidropsarus parini*	Chocolate-brown to grey	SE Pacific: Nazca Ridge; 310–680 m depth

## Data Availability

The sequences employed in the current study are available in the BOLD systems (https://www.boldsystems.org/, accessed on 27 April 2022) and GenBank (https://www.ncbi.nlm.nih.gov/genbank/, accessed on 27 April 2022) repositories. All specimens used in this study for taxonomical purposes are deposited in the fish collection of the Museo de Historia Natural, Universidade de Santiago de Compostela (MHNUSC) in Santiago de Compostela, Spain (see methods). All other data are included in this article.

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
