# Peer review of "Gaidropsarus gallaeciae (Gadiformes: Gaidropsaridae), a New Northeast Atlantic Rockling Fish, with Commentary on the Taxonomy of the Genus†"

_biology, 2022, doi:10.3390/biology11060860_

Round 1

Reviewer 1 Report

This manuscript is well-sound in general, and the new species seems to be adequately proved, with its morphology adequately described. I have only technical comments, and if the authors will agree to make these changes, this manuscript can be accepted without a new reconsideration.

Lines 29-31 (last sentence of simple summary) represent a very banal conclusion said nothing helpful.

Line 82: the reference(s) will be welcomed for the first sentence in this paragraph.

Section 3.3 (comparative material examined): species name needs to be entered only once for each species.

Line 217: Reference to Nelson et al. [27] is inadequate, as this is a compiled work with uncertain sources for the diagnoses provided. It will be better if the authors will provide here a reference to the diagnosis given in the taxonomic paper(s) (like Svetovidov (1986) or else).

Line 266: the limits of vertebral count for this complex and the opposed comparison group should be provided here.

Table 1: mean and its error are required for each measurement in % of SL & HL.

All figures are too small and should be enlarged as much as possible. Both radiographs (figs. 2 & 3) are of poor quality, at least they should be contrasted in a graphic editor. Fig. 4C should be rotated on 180 degree.

Author Response

Lines 29-31 (last sentence of simple summary) represent a very banal conclusion said nothing helpful.

R1: This paragraph has been deleted

Line 82: the reference(s) will be welcomed for the first sentence in this paragraph.

R2: A new reference (Morato et al., 2010) has been added

Section 3.3 (comparative material examined): species name needs to be entered only once for each species.

R3: This aspect has been corrected

Line 217: Reference to Nelson et al. [27] is inadequate, as this is a compiled work with uncertain sources for the diagnoses provided. It will be better if the authors will provide here a reference to the diagnosis given in the taxonomic paper(s) (like Svetovidov (1986) or else).

R4: We had selected Nelson because the recent consideration that the genus Gaidropsarus has three instead of two dorsal fins is reported in Nelson but not in Svetovidov. Following the recommendation, Nelson has been changed by Iwamoto and Cohen (2016) who also report three dorsal fins

Line 266: the limits of vertebral count for this complex and the opposed comparison group should be provided here.

R5: A new phrase has been rewritten accordingly

Table 1: mean and its error are required for each measurement in % of SL & HL.

R6: The mean and standard error have been added to table in a new column

All figures are too small and should be enlarged as much as possible. Both radiographs (figs. 2 & 3) are of poor quality, at least they should be contrasted in a graphic editor. Fig. 4C should be rotated on 180 degree.

R7: The figures 2 & 3 meet the publisher's minimum quality requirements; the fig 4C has been rotated

Reviewer 2 Report

Dear authors:

The manuscript is very good but it requires changes in methodology and in consequences in results:

1- About morphometric analysis 1: you must cite the at least one reference that support the morphometric parameters that you use.

2- About morphometric analysis 2: ideally for a good analysis, it is ideal that you have raw data of other species or population, and when you have at least  three data sets you must apply a multivariate analysis (in example discriminant analysis). 

General comments, once that you have the multivariate analysis based on morphometric parameters, you can compare these analysis with genetic analysis, and your manuscript will increase its quality.

Many success and blessings !!

Author Response

1- About morphometric analysis 1: you must cite the at least one reference that support the morphometric parameters that you use.

R1: A new reference has been added

2- About morphometric analysis 2: ideally for a good analysis, it is ideal that you have raw data of other species or population, and when you have at least three data sets you must apply a multivariate analysis (in example discriminant analysis). 

R2: We have tried but the number of available specimens was very low and some of the fundamental variables to differentiate by species, such as the number of vertebrae, are not available for all specimens, so the results were not sufficiently reliable.

Reviewer 3 Report

This article describes the finding and characterisation of a new deep-water fish species found on two different seamounts outside SW Europe. Gaidropsarus gallaeciae is the new species described, and the differences to other closely related species within the genus are described. Most of the others have more vertebrae, and also other key attributes differ. It appears that the present taxonomy of the genus is based on only a few specimens, due to difficulties with sampling, thus this contribution is welcome.

Some genetic characterisation is included, making it possible to construct a molecular cladogram separating the new species from the others.

I have no specific comments to improve the manuscript.

Author Response

There are not specific comments or corrections to this referee